# Microstructural Evolution and Strengthening Mechanism of SiC/Al Composites Fabricated by a Liquid-Pressing Process and Heat Treatment

**DOI:** 10.3390/ma12203374

**Published:** 2019-10-16

**Authors:** Sangmin Shin, Seungchan Cho, Donghyun Lee, Yangdo Kim, Sang-Bok Lee, Sang-Kwan Lee, Ilguk Jo

**Affiliations:** 1Composites Research Division, Korea Institute of Materials Science (KIMS), Changwon 51508, Korea; p996305s@kims.re.kr (S.S.); sccho@kims.re.kr (S.C.); gsprada@kims.re.kr (D.L.); gastank@hanmail.net (S.-B.L.); 2Materials Science and Engineering, Pusan National University, Busan 46241, Korea; yangdo@pusan.ac.kr; 3Department of Advanced Materials Engineering, Dong-Eui University, Busan 47340, Korea

**Keywords:** Al matrix composite, high volume fraction, liquid-pressing process, microstructural evolution, strengthening mechanism

## Abstract

Aluminum alloy (Al7075) composites reinforced with a high volume fraction of silicon carbide (SiC) were produced by a liquid-pressing process. The characterization of their microstructure showed that SiC particles corresponding to a volume fraction greater than 60% were uniformly distributed in the composite, and Mg_2_Si precipitates were present at the interface between the matrix and the reinforcement. A superior compressive strength (1130 MPa) was obtained by an effective load transfer to the hard ceramic particles. After solution heat treatment and artificial aging, the Mg_2_Si precipitates decomposed from rod-shaped large particles to smaller spherical particles, which led to an increase of the compressive strength by more than 200 MPa. The strengthening mechanism is discussed on the basis of the observed microstructural evolution.

## 1. Introduction

Aluminum matrix composites (AMCs) have many advantages, such as high strength, hardness, stiffness, modulus, and superior wear resistance; therefore, AMCs are considered to be attractive materials for various structural applications, such as the production of automotive parts and armor modules [1]. However, AMCs reinforced with a low volume fraction of ceramic particles have relatively poor mechanical properties and do not meet the requirements for those applications. To achieve the relevant mechanical properties, AMCs with a high volume fraction of silicon carbide (SiC) have been consistently developed and offer the double advantage of being lightweight and having excellent mechanical properties [2].

The mechanical properties of AMCs can be significantly improved by a few methods. By changing the reinforcement content in the composites, an enhanced load transfer effect from the increased volume of the reinforcement can be achieved [3]. The mechanical properties of a SiC/Al composite could be further enhanced if the grain structure of the metal matrix can be refined by incorporating small SiC reinforcements into the Al matrix. [4,5]. Furthermore, heat treatment improves the strength of the matrix alloy when composites are fabricated using heat-treatable Al alloys as a matrix, such as the 2xxx, 6xxx, and 7xxx series [6,7]. Wang et al. produced a high volume fraction of SiC-reinforced Al–Si–Mg alloy by pressureless infiltration and investigated the effect of heat treatment on its properties [8].

Several researchers have studied the strengthening mechanisms of low-volume-fraction SiC/Al composites [9,10]. Other researchers found that the dominant strengthening mechanisms in the composite were grain refinement, thermally induced dislocation, and dislocation pile-up at the interface [11,12,13]. In the case of AMCs reinforced with a high volume fraction of SiC, the main strengthening effect can be achieved through load transfer from the soft matrix to the hard ceramic particles [14]. However, investigation on the effect of the high-volume-fraction reinforcement on the SiC/Al composites has been limited. In addition, the matrix alloy fraction is comparatively low, and the heat treatment effect of the base metal is considered to be negligible.

The aim of the present study was to reveal the microstructure and mechanical properties of a high-volume-fraction SiC/Al7075 composite that contained a varying amount of reinforcement and is produced using a liquid-pressing process. The microstructural evolution following the T6 heat treatment and its effect on the mechanical properties of the final composite were investigated.

## 2. Experimental Procedure

### 2.1. Materials and Methods

SiC powders with commercial grades of F240 and F600 (FEPA-specified size distributions) were used as reinforcements (Saint-Gobain ceramic materials), and a T6 heat-treated Al7075 (Kaiser aluminum) alloy plate was used as a matrix to fabricate metal matrix composites. The mean particle size of the SiC reinforcements was analyzed with laser diffraction spectroscopy (Sympatech HELOS, Clausthal-Zellerfeld, Germany). Powder mixtures of the SiC reinforcements with 3:1 and 1:3 ratios were prepared in a 500 mL plastic bottle. The mixtures were then mixed for 1 h using a turbula mixer with a ball-to-powder mass ratio of 1:2 (10 mm, 5 g stainless steel ball). 

Table 1 shows the designation of each composite with the volume mean diameter (VMD) of the SiC reinforcement powder. The monomodal SiC powders with FEPA grit types of F600 and F240 had VMDs of 11.45 μm and 46.13 μm, respectively. The Al7075 alloy infiltrated with four types of reinforcements to produce the composites through the liquid-pressing process and the detailed composite designations are presented in Table 1. MS and BS mean monomodal SiC and bimodal SiC, respectively.

A liquid-pressing process was applied to carry out infiltration and produce the composite [15,16]. The Al7075 plates (50 mm × 50 mm × 2 mm) were placed into a steel mold (50 × 50 × 10mm), the SiC reinforcements were packed on the matrix, and then other Al7075 plates were placed on top of the packed SiC powder. After air evacuation, the mold assembly was heated to 850 °C, pressurized for 10 min with a 100 kg/cm^2^ load, furnace-cooled to 470 °C (solidus of Al7075), and finally air-cooled to room temperature. The composites and the unreinforced Al7075 alloy were solution-treated at 465 °C for an hour, followed by water quenching and aging at 120 °C for 24 h (T6 condition).

### 2.2. Characterization

ImageJ software version 1.52a was used to calculate the SiC volume fraction of each composite. This software generates histograms of SiC particles from the scanning electron microscope (SEM) image and quantifies the area of SiC which provides the volume fraction of all SiC reinforcements. The microstructures of the SiC powder andcomposites and the compressive fracture surface were observed with a SEM (SEM, LEO-1450, Zeiss, Oberkochen, Germany). The X-ray diffraction (XRD) patterns of matrix alloy and composite were measured by a Rigaku D/MAX-2500 diffractometer (Rigaku Co., Tokyo, Japan) with Cu K_α_ radiation (k = 0.15406 nm) operated at 40 kV and 100 mA. The interfacial morphology and detailed microstructure were characterized using transmission electron microscopy (TEM, JEM2100F, JEOL Ltd., Tokyo, Japan) and energy dispersive x-ray spectroscopy (EDS, X-Max 80T, Oxford, Abingdon, UK). The composition profiles of the composite structure were obtained from a quantitative electron probe microanalyzer (EPMA) on a JEOL JXA-8530F (Tokyo, Japan) microprobe with an accelerating voltage of 20 kV. 

The mechanical properties of the samples were evaluated by compression testing (Instron5882, Norwood, MA, USA). For each of the specimens, hardness values were determined by taking 20 measurements, and a statistical evaluation was performed. 

## 3. Results and Discussion

Figure 1 shows the microstructure of the composites with differently sized SiC and confirms that a SiC volume percentage above 60% was obtained in bimodal composites. As expected, the reinforcement volume fraction reached a maximum as the relative ratio of the two particles changed. The BS31 composite showed the maximum reinforcement volume fraction of 63%, a value about 20% higher than that obtained for MS24. This finding can be attributed to the increased packing density of the bimodal powder mixture. During the infiltration of the molten metal, the threshold pressure for infiltration was larger when using the small reinforcement particles, due to the smaller inter-particle distance. In the bimodal particle-reinforced Al, the permeability of Al decreases as the fine particle fraction and compactness of the powder increases [17]. Therefore, a high infiltration pressure is necessary for the infiltration of the molten Al in bimodal reinforcement to produce fewer defects in the composite. As the SEM micrograph shows, there were no major defects, such as agglomeration of SiC, pores, and non-infiltrated areas in the composites. These results indicate that the infiltration of the molten Al in the liquid-pressing process was quite effective and that a composite without defects and with a uniform distribution of SiC was successfully fabricated.

Figure 2 indicates the compression test results of the composites. The ultimate compressive strength (UCS) of the BS31 bimodal composite was found to be higher than that of BS13 (100 MPa after T6 heat treatment). The UCS of BS31 was about 300 MPa higher than that of the monomodal MS24 composite. The highest UCS was 1130 MPa, which is a high value for Al-based composites. Few studies report the compressive strength of high-volume-fraction SiC/Al composites under a quasi-static load. The UCS of the 50 vol.% SiC/Al2024 composite fabricated by squeeze casting [18,19] was reported as about 590 MPa and 530 MPa in two studies. The UCS of a 60 vol.% SiC-reinforced Al–Mg alloy produced through the pressure infiltration method was found to be about 570 MPa [20]. Thus, the composite fabricated by the liquid-pressing process was of good quality, and as a result, an effective load transfer from the matrix to the reinforcement occurred.

Also, it is clear that composites subjected to a precipitation-hardening heat treatment after annealing have higher UCS values, as compared to their respective as-infiltrated specimens. The UCS increased by about 200 MPa after T6 heat treatment. Typically, the heat treatment effect in the high volume fraction of reinforcement in ceramic-reinforced composites is unclear, due to the fact that an increase in the interfacial area between the reinforcement and the matrix may yield many defects [21]. However, in this study, the T6 heat treatment of the composites significantly increased the compressive strength of all specimens. These results clearly show the effect of heat treatment on AMCs reinforced with a high volume of SiC.

Figure 3 shows the XRD patterns of the BS31 composite and Al7075 matrix alloy that were produced by a liquid-pressing process, followed by annealing or T6 heat treatment. The main XRD peak of the BS31 composite corresponds to the Al matrix and SiC reinforcement. Generally, the precipitate-hardening phase of the Al7075 alloy by the T6 heat treatment process is known to produce an MgZn_2_ phase [22]. The formation of the MgZn_2_ phase was confirmed by the XRD analysis of the T6 heat-treated Al7075 matrix alloy. The inset with the low-intensity peak at 19–23° in the XRD spectrum of Al7075-T6 also confirms the presence of MgZn_2_ in the Al7075 alloy after T6 heat treatment. However, in the composite specimens, no MgZn_2_ phase was detected, but new peaks corresponding to the Mg_2_Si phase were observed, regardless of the heat treatment. The formation of these Mg_2_Si precipitates can be attributed to the diffusion that occurs between Mg from the Al7075 matrix and Si from the SiC particles during the high-temperature (850 °C) process. The standard Gibbs free energy of formation of Mg_2_Si is less than that of MgZn_2_ formation under the processing temperature; therefore, Mg_2_Si is thermodynamically preferred [23]. The absence of the MgZn_2_ phase might be due to the depletion of Mg to form Mg_2_Si during the solidification process, which could be confirmed by the XRD peaks of Zn shown in Figure 3. 

A Si phase peak with weak intensity was also detected for the composite, which revealed the dissolution of Si from SiC particles during the process. These Zn and Si phases were expected, due to the rapid solidification during the liquid-pressing process. According to the literature, Si is effective in restricting the formation of Al_4_C_3_ in SiC/Al composites [24]. In this composite, a dissolved Si phase could prevent the detrimental formation of Al_4_C_3_.

Figure 4a–d shows the transmission electron microscope (TEM) images and corresponding energy dispersive x-ray spectroscopy (EDS) elemental mapping of the composite after annealing heat treatment. Figure 4a shows that a rod-like interfacial phase, marked with yellow dots, existed at the interface between the Al7075 matrix and the SiC reinforcement. The Mg–Si map revealed this phase as Mg_2_Si along the interface with SiC, which is in agreement with the electron backscatter diffraction (EBSD) and XRD results. According to the results of microstructural analysis and the Al–Mg–Si phase diagram [25,26], the solidification process in this composite can be described by Equation (1). The subscripts p and e represent the primary and eutectic phases, respectively. During the high-temperature liquid-pressing process, Si dissolved from the SiC particles might act as a substrate for primary Mg_2_Si nucleation, and therefore, a primary Mg_2_Si precipitate may occur at the surface of the SiC particles. Then, this Mg_2_Si intermetallic phase further acts as a heterogeneous site for the nucleation of α-Al, which lowers the interfacial energy. As a result, the Mg_2_Si phase is covered by a layer of Al–Mg_2_Si in a binary eutectic structure (marked as f on Figure 4e).

L → L_1_ + Mg_2_Si_p_ → (α-Al + Mg_2_Si)_e_ + Mg_2_Si_p_(1)

In order to analyze the interfaces across the Al7075 matrix, Al–Mg_2_Si eutectic phase, and SiC reinforcement, high-resolution TEM images were acquired at the interfacial areas, as shown in Figure 4f,g that represents the areas marked in blue (f) and red (g) in Figure 4e. As Figure 4f indicates, examination across the interfaces confirmed that the interfaces were clean and there was no layer of contamination or defects. These interfaces formed during the solidification process resulted in a strong bond between the Al matrix and its SiC reinforcement. Also, well-distributed hard Mg_2_Si with strong interfacial bonding, which is a result of the heat treatment, increased the mechanical properties of the composite. Insets in Figure 4g show the fast Fourier transforms (FFTs) images of the Mg_2_Si precipitate and the selected area diffraction pattern (SADP) of SiC particles. Note that the SADP and FFTs of the regions highlighted in the SiC and Mg_2_Si phases only show the set of spots that correspond to the 6H–SiC phase oriented to the [11$\bar{2}$0] zone axis and to the cubic Mg_2_Si phase along the [$\bar{1}$01] zone axis, respectively.

Figure 5a–d shows the EPMA analysis results, including the distribution of Si, Mg, and Zn elements in the selected region of the T6 heat-treated BS31 composite. As shown in Figure 5b, dissolved elemental Si was observed along the SiC particle boundary in the T6 heat-treated specimen. Elemental mapping of the images also displayed the uniform distribution of carbon in the Al matrix, showing that there were no significant differences of carbon before and after heat treatment. The EPMA mapping image clearly shows the uniform distribution of the Zn alloying element throughout the matrix phase region as a solid solution (Figure 5c). The mapping of Mg (Figure 5d) shows the segregation of a submicron (~1 μm)-sized spherical Mg-rich phase at the interface (yellow arrows) between the matrix and the reinforcement. From the XRD and EPMA maps, these Mg-rich phases are clearly Mg_2_Si precipitates. The EPMA analysis results of the as-annealed composite are shown in Figure 5e,f. As compared to the T6 heat-treated specimens, the as-annealed samples contained 1–5 μm sized large phases of spherical and rod-shaped particles, which are indicated by dotted white circles and yellow arrows (Figure 5f). From these results, it could be deduced that the strengthening effect of the heat treatment on the composite derived from the formation of a small and spherical Mg_2_Si phase in the T6 heat-treated specimens. After solution treatment, the rod-shaped large Mg_2_Si phase precipitated from the supersaturated solid solution; then, these phases decomposed into smaller spherical Mg_2_Si particles during the aging treatment. [27] The change of Mg_2_Si morphology in the composite with the aging process resulted from the need to minimize the energy of the system in metastable quenched conditions.

In this research, Al7075 composites with a high volume fraction of SiC reinforcement were successfully produced through the liquid-pressing process. The interface created during the solidification process was clean without defects. It is considered that the improvement of the mechanical properties of composite, as compared to those of an unreinforced matrix, was mainly due to an efficient load transfer from the Al7075 matrix to the SiC reinforcement. In addition, the uniform distribution of the hard spherical Mg_2_Si phase (elastic modulus = 120 GPa) at the interface after T6 heat treatment further contributed to the strengthening of the SiC/Al composite.

## 4. Conclusion

In summary, Al7075 matrix composites that contain monomodal- and bimodal-sized SiC reinforcements were fabricated by a liquid-pressing process. The SiC reinforcement was distributed uniformly in the matrix and presented a clean interface without major defects and with good bonding to the matrix. The incorporation of SiC and the control of the reinforcement volume fraction significantly improved the mechanical properties of the composites through effective load transfer from the matrix to the reinforcement, because of good interfacial bonding. The bimodal composite with a 3:1 particle ratio after T6 heat treatment exhibited the maximum ultimate compressive strength (1130 MPa). Spherical Mg_2_Si precipitates were largely homogeneously dispersed in the Al7075 matrix during the process. These Mg_2_Si phases further decomposed into smaller spherical Mg_2_Si during the precipitation-hardening heat treatment. From these results, it may be concluded that the strengthening of SiC/Al7075 composites that are fabricated by a liquid-pressing process can be attributed to the effective load transfer to the SiC reinforcement, as well as to the uniform distribution of hard spherical Mg_2_Si precipitates at the interface.

## Figures and Tables

**Figure 1 materials-12-03374-f001:**
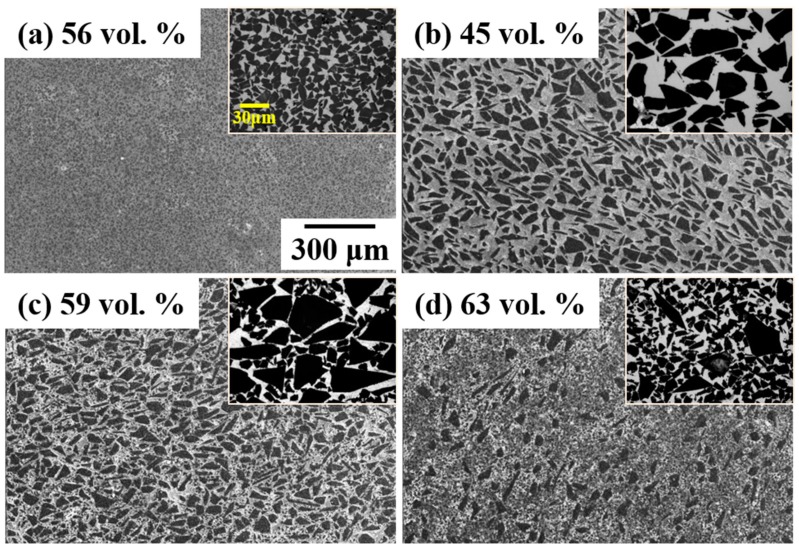
Scanning electron microscope (SEM) images of the (**a**) MS60; (**b**) MS24; (**c**) BS13; and (**d**) BS31 composites with a calculated SiC volume fraction (at the same magnification).

**Figure 2 materials-12-03374-f002:**
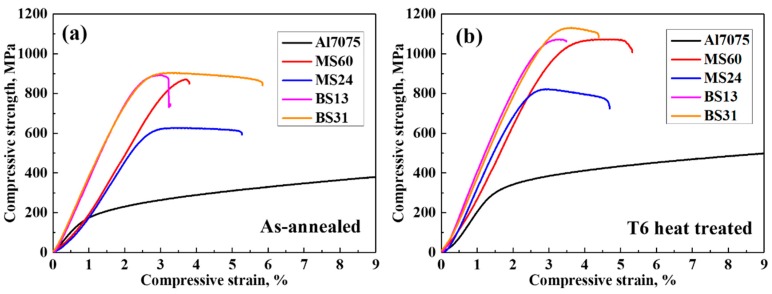
Compressive strength of the composites in (**a**) as-annealed and (**b**) T6 heat-treated states.

**Figure 3 materials-12-03374-f003:**
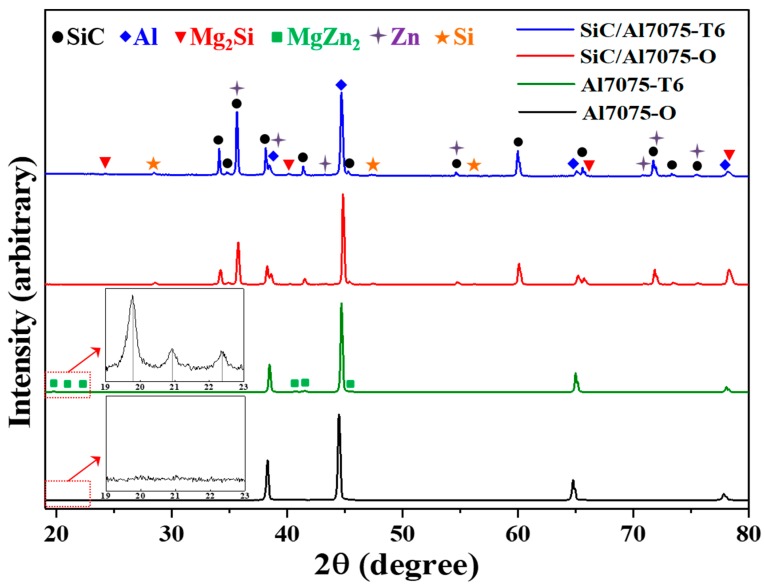
The X-ray diffraction pattern of the BS31 composite and Al7075 matrix after annealing or T6 heat treatment (insets: enlargement of the 19°–23° region).

**Figure 4 materials-12-03374-f004:**
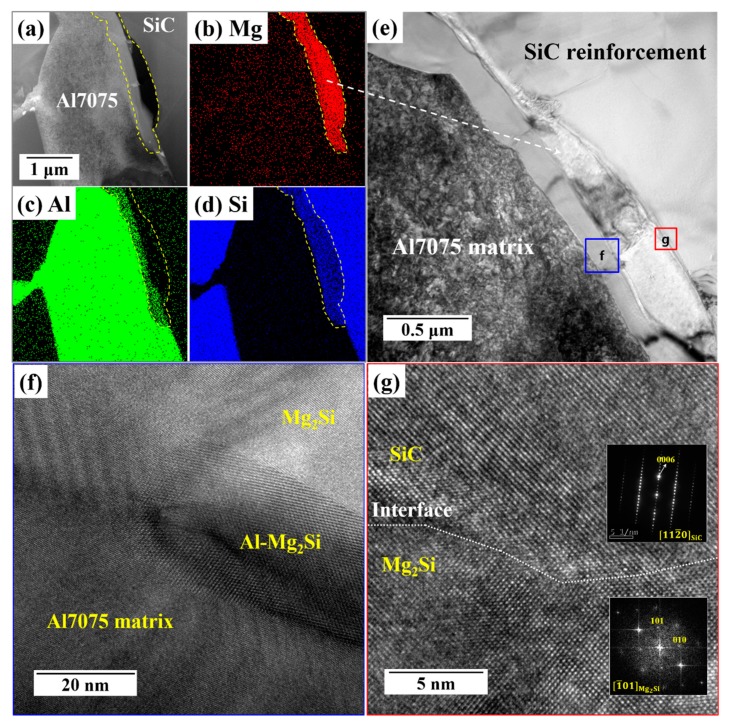
Transmission electron microscope (TEM) images of the BS31 composite after annealing: (**a**–**d**) TEM image and EDS mapping; (**e**–**g**) high-resolution TEM morphology of the interface between the Al7075 matrix, the Mg_2_Si precipitates, and the SiC reinforcement. The selected area diffraction pattern (SADP) and fast Fourier transform pattern correspond to the 6H–SiC and cubic Mg_2_Si phases, respectively.

**Figure 5 materials-12-03374-f005:**
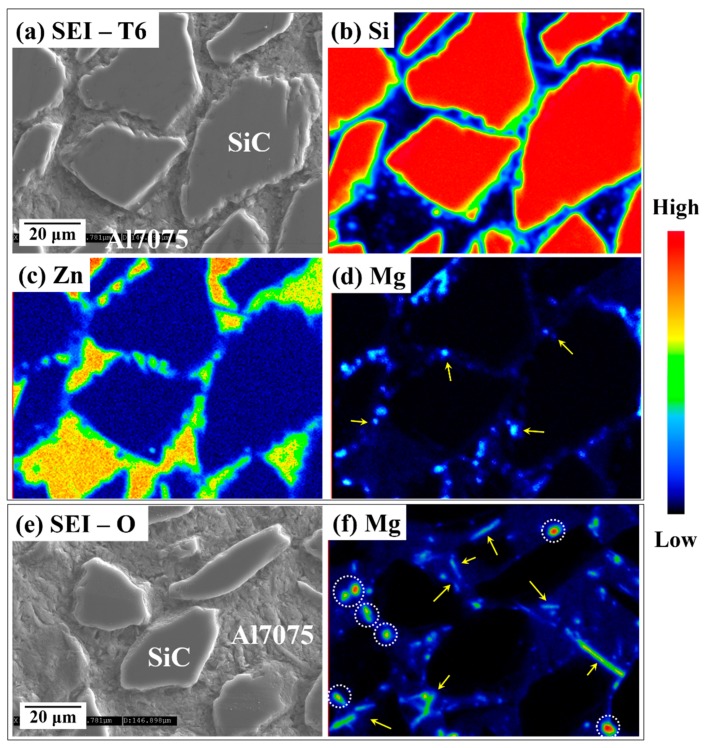
(**a**) Secondary electron imaging (SEI) of the MS24 composite after T6 heat treatment; (**b**–**d**) EPMA analysis of the element distribution map of the area in (**a**); (**e**) SEM image of as-annealed MS24; and (**f**) Mg element map.

**Table 1 materials-12-03374-t001:** Content of the silicon carbide (SiC) reinforcement and the designation of the composites: V_f_ is the particle volume fraction, and VMD is the reinforcement volume mean diameter; F240 and F600 refer to the commercial grades of SiC powders; MS and BS mean monomodal SiC and bimodal SiC, respectively.

Reinforcement Type	Composite Designation			
	MS60	MS24	BS31	BS13
F600 V_f_ (VMD = 11.45 μm)	100	-	75	25
F240 V_f_ (VMD = 46.13 μm)	-	100	25	75

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
