# Peer review of "Microstructural Evolution and Strengthening Mechanism of SiC/Al Composites Fabricated by a Liquid-Pressing Process and Heat Treatment"

_materials, 2019, doi:10.3390/ma12203374_

Round 1

Reviewer 1 Report

The authors need to revise the submitted work titled "Microstructural evolution and strengthening mechanism of SiC/Al composites fabricated by a liquid pressing process and heat treatment" to be considered for publication, and the following are the reviewer comments:

The Introduction section is very short. There are repeated sentences on the load transfer mechanism (Lines 35&43). Aluminium being a low melting point metal, previously, there are several works on Al-SiC(and other ceramic) metal matrix composites fabricated via powder metallurgy and liquid metal infiltration processes. The authors need to elaborate more on the motivation of the research and highlight the key findings of the previous work. Also Al alloy selection (Al7075) needs to be justified.  What is the need for the bi-modal SiC particles instead of mono-sized SiC as the starting material?  Elaborate more on the packing of SiC and Al matrix (Line 68). If the SiC and Al are packed, how can it be a infiltration process? Line 75, please highlight the imageJ software version and tools used in the software. How was the volume fraction calculated? Line 91, what is BS31 composite? Why suddenly this term is introduced? Define the composite in the introduction section. Lines 108-114, tabulate the previous results from the scientific community and compare the compression properties to that of the fabricated composites. There are several other Al-SiC based composites in the literature. Also, compare the results of low and high volume fraction SiC reinforced Al composites.

Author Response

Thank you for the detailed suggestions and comments to our manuscript. These reviews have been very helpful in enhancing our paper. Below is a detailed discussion of the changes we have made to address the reviewer comments.

The Introduction section is very short. There are repeated sentences on the load transfer mechanism (Lines 35&43). Aluminium being a low melting point metal, previously, there are several works on Al-SiC(and other ceramic) metal matrix composites fabricated via powder metallurgy and liquid metal infiltration processes. The authors need to elaborate more on the motivation of the research and highlight the key findings of the previous work.

→ The authors carefully revised the introduction section accordingly. As the reviewer suggested, the authors did a literature survey again and added 6 new references, including studies on the SiC/Al composites which fabricated by other processes.

Also Al alloy selection (Al7075) needs to be justified. 

→ We supplemented the contents including detailed information of type and size of used Al7075 plate.

What is the need for the bi-modal SiC particles instead of mono-sized SiC as the starting material?

→ The reason why we use bi-modal SiC reinforcement is to increase the volume fraction of SiC with increasing the packing density. With increase in the SiC volume fraction, composites’ mechanical properties could be improved. We indicate this contents in the introduction section.

Elaborate more on the packing of SiC and Al matrix (Line 68). If the SiC and Al are packed, how can it be a infiltration process?

→ In this research, the form of Al7075 matrix was not powder but plate. We apologize for any confusion. As the reviewer pointed out, preparation of reinforcement and matrix are specifically revised.

Line 75, please highlight the imageJ software version and tools used in the software. How was the volume fraction calculated?

→ We revised the ‘2.2. Characterization’ section including additional information on the imageJ software and how we calculate the volume fraction of SiC reinforcement as commented.

Line 91, what is BS31 composite? Why suddenly this term is introduced? Define the composite in the introduction section.

→ The designation of all composites are indicated in table 1. Also, we add the explanatory note on the designation of composite in the’ 2.1. Materials and method’ section as commented by the reviewer.

Lines 108-114, tabulate the previous results from the scientific community and compare the compression properties to that of the fabricated composites. There are several other Al-SiC based composites in the literature. Also, compare the results of low and high volume fraction SiC reinforced Al composites.

→ As the reviewer pointed out, we have tried to find more research on the compressive strength of composite with low or high volume fraction of SiC reinforced Al matrix. Unfortunately, it is hard to find articles other than references we cited in our paper.

Reviewer 2 Report

The aim of the paper, i.e. the investigation of the effect of the high volume fraction reinforcement on the SiC/Al composites, is quite interesting. The aim of this paper is to reveal the microstructure and mechanical properties of a high volume-fraction SiC/Al7075 composite that contains a varying amount of reinforcement and is produced using a liquid pressing process. The microstructural evolution from the T6 heat treatment and its effect on the mechanical properties of the final composite were also investigated.

The abstract summarize the work. The purpose of the study is clearly outlined and the findings of prior work are well discussed. There are no errors in logic or experimental procedure. The authors accurately explain how the data were collected. There is sufficient information that the experiment can be reproduced. All topics are well presented and discussed. The summary and conclusions are sound and justified. All presented figures are good quality and they prove their point. The paper is written in good English. The manuscript is easily readable concerning language, style and presentation. The references are appropriate and up to date.

Author Response

We truly appreciate all the constructive and positive comments.

Reviewer 3 Report

This paper deals with Al-SiC composites with high volume fractions of the reinforcement. The processing, structural features and mechanical properties of the composites are reported. I think that these results are valuable as, at present, not so many studies are conducted on metal matrix composites with high volume fraction of reinforcements. It is important to understand the strengthening mechanisms operating in such composites. 

I have some suggestions to improve the paper.

Please explain the designation of the composites - letters/numbers used (Table 1).

Why is the term "liquid pressing" used? It appears that "infiltration" would be a better term.

Please clarify what Al 7075 alloy was selected as a matrix.

What was the purpose of making composites using a mixture of two SiC powders? Please explain your idea in the Introduction.

I think that "matrix" is not a suitable term for these composites as the content of the metal is not high and the metallic phase is no longer the dominant one. Please consider using "binder" instead of "matrix".

If Mg2Si forms from silicon contained in SiC, to which phase does the remaining carbon go?

I do not think that from the XRD pattern that you show you can conclude on the presence of free Si and Zn. Please clarify and revise the corresponding discussion. Lines 140-142 are not clear with regard to the behavior of Zn.

Author Response

Thank you for compliments and comments to the manuscript. Your sincere reviews have been very helpful to our research. Below is our reply to your comments.

Please explain the designation of the composites - letters/numbers used (Table 1).

→ The designation of all composites are indicated in table 1. Also, we add the explanatory note on the designation of composite in the’ 2.1. Materials and method’ section as commented by the reviewer.

Why is the term "liquid pressing" used? It appears that "infiltration" would be a better term.

→ We entirely agree. During the typical infiltration process, SiC ceramic preform packed by mechanical press is used. But in the liquid pressing process we do not make the SiC preform. We press the steel mold using press after melting of the Al matrix into the SiC powder. More details can be found from the references we cited in the introduction section.

Please clarify what Al 7075 alloy was selected as a matrix.

→ We supplemented the contents including detailed information of type and size of used Al7075 plate.

What was the purpose of making composites using a mixture of two SiC powders? Please explain your idea in the Introduction.

→ The reason why we used bi-modal SiC reinforcement is to increase the volume fraction of SiC with increasing the packing density. With increase in the SiC volume fraction, composites’ mechanical properties could be improved. We indicate this in the introduction section.

I think that "matrix" is not a suitable term for these composites as the content of the metal is not high and the metallic phase is no longer the dominant one. Please consider using "binder" instead of "matrix".

→ In the composite terminology, continuous phase is described as the matrix. So we indicate the Al alloy as the ‘matrix’ to avoid any confusion.

If Mg2Si forms from silicon contained in SiC, to which phase does the remaining carbon go?

→ In the XRD results, possible peaks of carbon are overlap with other phases peaks. From the EPMA analysis, we confirmed that the free carbon exists in the matrix with small amount. They are uniformly distributed in the Al matrix. Similar results can be also found from the article ‘Ceramics International 43 (2017) 1755–1761’. We did not add carbon EPMA image since there is no significant meaning or differences before and after heat treatment. As the reviewer pointed out, we insert explanatory note about carbon.

I do not think that from the XRD pattern that you show you can conclude on the presence of free Si and Zn. Please clarify and revise the corresponding discussion. Lines 140-142 are not clear with regard to the behavior of Zn.

→ We truly appreciate the constructive comments. The authors claim that the existence of free Zn could be concluded from the coupling of XRD and EPMA analysis results. In case of free Si, many research on the SiC/Al composite argue the existence of free Si and what is reason for that. As confirm from the many researches, at high temperature, the SiC/Al system is instable and SiC particle tend to dissolve in the molten Al. So we suppose that Si is dissolved in the melted Al mixed with the solid SiC particles and somehow it remains after the rapid solidification during the liquid pressing process. More detailed mechanism of free Si and how it affects the composite is further to be investigated as the reviewer commented.

Round 2

Reviewer 1 Report

The authors have carefully considered the reviewers' comments and have revised the manuscript and the submitted manuscript can be considered for publication.

Author Response

The authors are thankful for the detailed suggestions and comments to our manuscript again.

Reviewer 3 Report

The authors have worked on the paper and improved its quality substantially.

My concern is still about the presence of free Zn as phase. It is marked on the XRD pattern as "Zn", while the authors say that Zn is highly soluble in Al. Then Al(Zn) solid solution should be formed. Also, I do not think there is enough evidence of the presence of free Si as a phase.

Author Response

→ Thanks for the delicate comment. We entirely agree with the reviewer’s comment. So we revised confusing sentences regarding existences of Zn phase and Zn solubility in Al matrix. The solubility of Zn is high in the Al, but the rapid solidification process during the liquid pressing process yield Zn phase solved in the Al matrix without forming solid solution.

Other possibility is formation of intermetallic compounds from Al-Mg-Si-Zn-C system such as Mg3Zn3Al2, Al4Si2C5, Al8SiC7, and dissolution of IMC phases into matrix followed by supersaturation of the Al with Zn, Si solute atoms.

From the current data, it is hard to decide exact status of these phases. But the authors are want to focus on the formation and evolution of the Mg2Si phase which effect mostly to the mechanical properties of the composite.

It would be very helpful for understanding quantitative analysis of element to search inter metallic compounds (IMCs) in the composites by using XPS as commented by reviewer. We hope to investigate detailed IMCs phase using XPS for the future work and submit other manuscript to the ‘Materials’ in the near future.
